# Language decoding from human brain activity via contrastive learning

**Matteo Ferrante**
Department of Biomedicine and Prevention
University of Rome, Tor Vergata
`matteo.ferrante@uniroma2.it`

**Nicola Toschi**
Department of Biomedicine and Prevention
University of Rome, Tor Vergata
Martinos Center For Biomedical Imaging
MGH and Harvard Medical School (USA)

**Alexander Huth**
Department of Computer Science & Neuroscience
UT Austin (USA)

## Abstract

We propose a novel contrastive learning approach to decode brain activity into sentences by mapping fMRI recordings and text embeddings into a shared representational space. Using data from three subjects, we trained a cross-subject fMRI encoder and demonstrated effective sentence identification with a retrieval module. Our model shows strong alignment between brain activity and linguistic features, with top-1 accuracy up to 49.2% and top-10 accuracy up to 84%, significantly outperforming chance levels. Our results highlight the potential of contrastive learning for cross-subject language decoding,

## 1   Introduction

Language is one of the most ubiquitous ways we experience the flow of information in our daily lives. We read, speak, communicate, and even think through language. It is a complex phenomenon that allows us to understand and convey information to others. To do this, our brain generates semantic representations of everything we encounter, taking context into account. Recent research has demonstrated a convergence between brain activity [3, 2, 9] during language tasks, such as listening or reading, and the way large language models process sentences. This has been shown through brain recordings using both non-invasive methods, such as fMRI, EEG, and MEG, as well as invasive techniques like LFP and ECoG. Powerful encoding and decoding models have been developed to link external stimuli, such as acoustically perceived sentences, images, videos, music with neural representations recorded during these tasks [19, 1, 6, 17, 4, 18, 13, 15, 7, 8, 14, 16]. These models have shown remarkable results across both invasive and non-invasive brain recording techniques. Here, we propose a novel method based on contrastive learning to learn a cross-subject fMRI encoder that projects fMRI recordings and text embeddings into a shared space, enabling sentence identification with a retrieval module. Our model takes as input fMRI activity and computes fMRI embeddings that can be compared with pre-computed sentence embeddings belonging to a set of candidates. By selecting the closest sentences in this space we can effectively decode the brain activity and obtain clues about semantic representation in the brain. Fig. 1 show a scheme of the pipeline proposed here.

38th Conference on Neural Information Processing Systems (NeurIPS 2024).

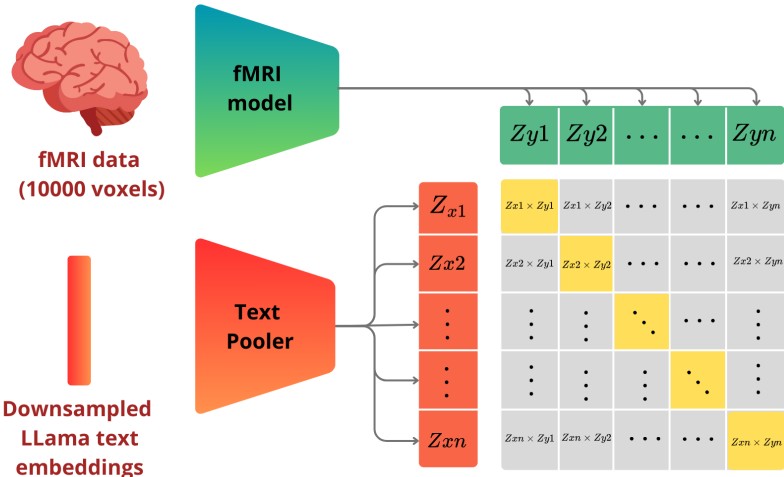

Figure 1: Schematic representation of the contrastive learning pipeline. The fMRI model encodes brain activity from 10,000 voxels located in the cortex into a shared latent space, while the text pooler processes downsampled LLama text embeddings. Pairwise dot products between fMRI and text embeddings are computed to create a similarity matrix, which is used for aligning brain activity with corresponding linguistic features. Yellow cells highlight correct matches between fMRI and text embeddings.

## 2 Material and Methods

### 2.1 Data

In our analysis, we used the publicly available dataset from [12]. We focused on subjects S1, S2, and S3, each of whom underwent fMRI recordings for approximately 16 hours while listening to 83 stories from The Moth and Modern Love podcasts. The fMRI data was collected using a 3T Siemens Skyra scanner with a repetition time (TR) of 2.00 seconds and an isotropic voxel size of 2.6 mm. This allowed for the measurement of BOLD signals, which reflect neural activity with an inherent delay due to the hemodynamic response. Preprocessing steps included motion correction, cross-run alignment, standardization and detrending of low-frequency. For more details, we refer the reader to the original paper [12]. Data from the first 70 stories of each subject were used as training set, while the other 12 stories were used as validation set. Also, the story *"wheretheressmoke"*, which was listened to 10 times to ensure better signal to noise ratio at test set, aligning with recent literature on language encoding and decoding [19, 1].

### 2.2 Encoding

The first step of our pipeline involves reducing the complexity of the signal we need to process. While much of the brain is active during language and semantic tasks [10, 11], certain regions are more directly involved in language processing. Therefore, we begin by identifying cortical regions that, at the voxels level, are more easily modeled by a language model, allowing us to focus our analysis on these regions. We trained an encoding model of brain activity using a large language model (LLM) as the foundation. For each word in the training stories, we computed embeddings from LLama3-8B [5], specifically from the 13th layer, using a context window of the previous five words. This choice of layer is supported by previous studies [1] that found early layers in LLama models exhibit stronger correlations with brain activity. To align the word embeddings with the fMRI temporal resolution, we downsampled them using a Lanczos filter, creating matching sentence-fMRI pairs. Finally, we calculated the Pearson correlation between the predicted and true brain activity on held-out validation data, selecting the top 10,000 cortical voxels as our target regions. The activity in these voxels are the targets we aim to embed and decode.

## 2.3 Contrastive fMRI model

The core contribution of this work lies in the proposed representation learning pipeline. Since a single TR of fMRI recording can be influenced by several preceding words due to the hemodynamic response (HRF), we must account for this effect. To address the unique properties of fMRI data, we developed a model consisting of two neural networks: an fMRI encoder and a text embedding pooler. The fMRI model is a cross-subject neural network. The first layer is subject-specific and projects the top 10,000 acivity corresponding to the top 10000 voxels of each subject into a shared representation space of dimensionality *common_dim*. The remaining layers of the network are shared across subjects, consisting of a multi-layer perceptron (MLP) that transforms the data from *common_dim* to the final latent dimension *latent_dim*. The text embedding pooler is an MLP that processes four downsampled word embeddings corresponding to the TRs that might affect the measured brain activity (i.e. those from 1, 2, 3, and 4 timepoints before the corresponding BOLD response). Each word embedding is projected into the final latent space by a linear layer that reduces the dimensionality. These projections are then concatenated, resulting in a final sentence representation of dimensionality *latent_dim*. In appendix, we detail all the hyperparameter search and architectures. Let $x$ represent the fMRI activity, $y$ the downsampled word embeddings, $f$ the fMRI model, and $g$ the text pooler. We define the projections as: $z_x = f(x)$, $z_y = g(y)$. Let $z_x \in \mathbb{R}^{n \times d}$ represent the encoded fMRI features and $z_y \in \mathbb{R}^{n \times d}$ represent the encoded text embeddings, where $n$ is the batch size and $d$ is the latent dimensionality. The logits matrix, which contains the similarity scores between each pair of fMRI and text embeddings, is computed as:

$$\text{logits}_{ij} = \frac{z_{x_i} \cdot z_{y_j}^\top}{\tau}$$

where $\tau$ is the temperature parameter that controls the sharpness of the distribution. The pairwise similarities are computed using the dot product between the fMRI and text embeddings: $\text{similarities}_{ij} = z_{x_i} \cdot z_{y_j}^\top$. Let the target labels be the identity mapping, where each input is matched with itself in the contrastive learning task. The targets vector $t \in \mathbb{N}^n$ is defined as: $t = \{0, 1, \ldots, n-1\}$ This implies that $t_i = i$, ensuring each fMRI sample is matched with the corresponding text sample. The contrastive loss $\mathcal{L}$ is computed as a combination of two cross-entropy losses, one for the alignment of fMRI to text and the other for text to fMRI:

$$\mathcal{L} = \frac{1}{2} \left( \mathcal{L}_{\text{CE}}(\text{logits}, t) + \mathcal{L}_{\text{CE}}(\text{logits}^\top, t) \right)$$

where $\mathcal{L}_{\text{CE}}$ denotes the cross-entropy loss function. This formulation optimizes both directions in the contrastive learning objective, ensuring that fMRI features are closely aligned with the corresponding text embeddings and vice versa. This contrastive learning approach ensures that both fMRI activity and the corresponding text embeddings are projected into the same latent space, aligning brain activity with linguistic features.

## 2.4 Retrieval and Evaluation

The retrieval module compares the L2 distance between fMRI and text embeddings across the test set to find the closest sentences to each fMRI sample. We evaluate the decoded sentences using three metrics: identification accuracy, top-1 accuracy, and top-10 accuracy. For top-10 retrieval, we select the 10 closest sentences to each fMRI TR based on L2 distance, with the target sentence defined as the previous 4 TRs (8 seconds) plus 5 preceding context words. Identification accuracy, adapted from vision and music decoding literature, measures how well the model identifies the correct sentence by comparing self-correlations in the latent space with other correlations. We compute Pearson correlations between the predicted vectors and targets, storing the results in a correlation matrix, and successively calculate identification accuracy by comparing the self-correlation with others in the same row and normalizing the result.

# 3 Results

The results from our contrastive learning-based language decoding model are shown in Figure 2 and Table 3. The top panel of Figure 2 illustrates the model's encoding performance, with Pearson

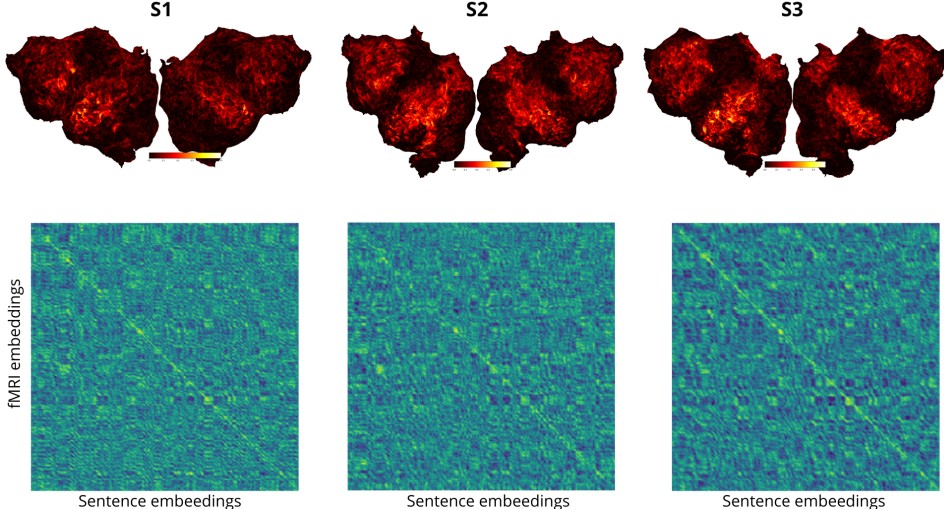

Figure 2: Top: Pearson correlation maps of predicted vs. actual brain embeddings for subjects S1, S2, and S3 shown on flattened cortical surfaces. The regions we identified are typically associated with language processing, such as the superior temporal gyrus and parts of the inferior frontal cortex. Bottom: Cosine similarity matrices between sentence embeddings and brain embeddings for each subject.

correlations between predicted and actual brain embeddings across three subjects (S1, S2, S3). The red regions indicate strong correlations, suggesting the model captures language-related cortical activity patterns, likely in language-processing areas of the brain. The consistent performance across subjects highlights the model's robustness in identifying key neural features associated with sentence comprehension. In the bottom panel, the cosine similarity matrices between sentence embeddings and brain embeddings show a strong alignment, indicated by bright diagonal lines, reflecting the model's ability to map linguistic structures to brain representations effectively. Table 3 provides quantitative metrics for decoding accuracy. The Top-1 Accuracy, which ranges from 0.313 (S2) to 0.498 (S3), significantly outperforms chance levels, confirming the model's ability to predict precise sentences. Top-10 Accuracy further validates this, with values as high as 0.838 (S3), indicating that the correct sentence is frequently among the top 10 predictions. Identification Accuracy is also high for all subjects, ranging from 0.910 (S2) to 0.962 (S3), reinforcing the model's strong performance in decoding brain representations of sentences. Overall, both Figure 2 and Table 3 demonstrate the model's effectiveness in linking sentence embeddings to brain activity, with strong performance across subjects. Subject S3 consistently shows the best results, suggesting individual differences in brain activity may influence decoding accuracy, offering avenues for future investigation.

| Subject | Top-1 Acc | Top-10 Acc | Chance Level Top-1 | Chance Level Top-10 | Identification Acc |
|---------|-----------|------------|--------------------|---------------------|---------------------|
| s1 | 0.3780 | 0.786 | 0.0114 | 0.0894 | 0.9571 |
| s2 | 0.3127 | 0.6666 | 0.0116 | 0.0855 | 0.9100 |
| s3 | 0.4982 | 0.8381 | 0.0118 | 0.0814 | 0.9624 |

Table 1: Performance metrics for Top-1, Top-10, and Identification Accuracy across different datasets.

## 4 Discussion and Conclusions

Contrastive learning has proven to be a robust method for learning cross-subject mappings between brain activity and sentence-level embeddings. However, a key limitation of our approach is that decoding is performed through a retrieval module (i.e., sentence identification). This requires access to candidate sentences beforehand, limiting the model's ability to generalize to brain activity related to sentences that differ significantly from those in the training dataset. Another important limitation pertains to future applications of this work. The learned fMRI embeddings could potentially be

used in conjunction with Bayesian decoding techniques or as inputs to modified large language models (LLMs) for open vocabulary decoding. However, these approaches raise concerns about privacy and bias. It will be crucial to address how to distinguish between actual thoughts and brain representations, and how to prevent biases in both models and human interpretations from influencing the results. Future research should explore the concept of neural privacy and develop strategies to disentangle model biases from genuine cognitive processes. In conclusion, this work presents a cross-subject architecture that decodes brain activity into sentences using contrastive learning and sentence identification, laying the groundwork for future advancements in brain-to-language decoding.

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

# A   Appendix / supplemental material

## A.1   Neural Network Architectures

This appendix describes the architecture of the neural networks used in this work. The models consist of an `Encoder` for processing the input data, an `Embedding Pooler` for projecting embeddings to a common latent space, and a `Contrastive Model` for learning via contrastive losses.

### A.1.1   Encoder

The `Encoder` network is designed to map the input data to a lower-dimensional latent space. The architecture is summarized in Table 2. $input\_dim$ is set to be the number of cortical voxels (10000) while $common\_dim$ is chose to be 4096.

Table 2: Encoder Network Architecture

| Layer Type | Dimensions | Activation Function |
|---|---|---|
| Input Layer | $input\_dim \times hidden\_dim$ | - |
| Alignment Layer (key = 1) | $input\_dim \times common\_dim$ | - |
| Alignment Layer (key = 2) | $input\_dim \times common\_dim$ | - |
| Alignment Layer (key = 3) | $input\_dim \times common\_dim$ | - |
| Layer Normalization | $common\_dim$ | - |
| Linear Layer | $common\_dim \times hidden\_dim$ | Identity |
| Linear Layer | $hidden\_dim \times latent\_dim$ | - |
| Layer Normalization | $latent\_dim$ | - |

The alignment layers apply different transformations to subsets of the data depending on key values provided at runtime. These layers are followed by sequential linear layers with ReLU activation.

### A.1.2   Embedding Pooler

The `Embedding Pooler` projects embeddings, such as fMRI data, into a lower-dimensional latent space. The architecture is summarized in Table 3.

Table 3: Embedding Pooler Architecture

| Layer Type | Dimensions | Activation Function |
|---|---|---|
| Input Layer | $input\_dim \times hidden\_dim$ | - |
| Layer Normalization | $input\_dim$ | - |
| Linear Layer | $input\_dim \times hidden\_dim$ | GELU |
| Linear Layer | $hidden\_dim \times (latent\_dim/4)$ | GELU |
| Layer Normalization | $(latent\_dim/4)$ | - |
| Reshaping | $(batch\_size, -1)$ | - |

The input is first normalized, followed by a sequence of linear layers and GELU activation. The final step reshapes the output into a pooled embedding vector.

### A.1.3   Contrastive Model

The `Contrastive Model` combines the `Encoder` and `Embedding Pooler` for multimodal learning using contrastive loss functions. The model supports various loss functions, such as contrastive loss, mean squared error (MSE), and cosine similarity loss. The overall architecture is shown in Table 4.

Table 4: Contrastive Model Components

| Component | Description |
|---|---|
| Encoder | See Table 2 |
| Embedding Pooler | See Table 3 |
| Loss Function | Contrastive |

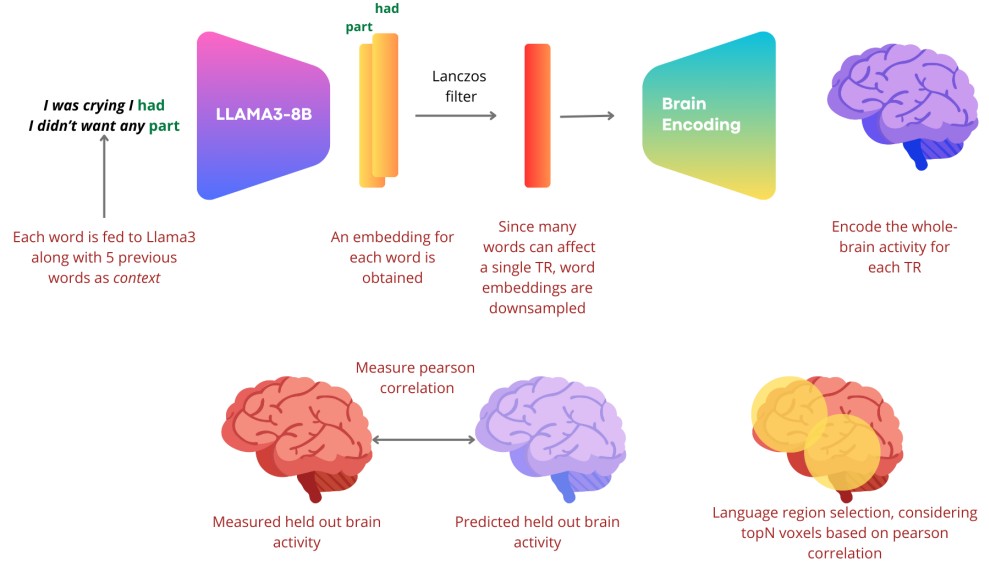

Figure 3: Encoding scheme

During training, the model minimizes the distance between fMRI projections and embeddings from other modalities. The training process is logged and monitored using the mean squared error, cosine similarity, and the primary loss function.

## A.2  Training Setup

The models are trained using an AdamW optimizer with a learning rate of $1e - 4$ and weight decay of $1e - 4$. The learning rate is adjusted dynamically using a scheduler that reduces the rate by a factor of 0.1 when the validation loss plateaus for 50 epochs.

Table 5: Training Parameters

| Parameter | Value |
|---|---|
| Optimizer | AdamW |
| Learning Rate | $1e - 4$ |
| Weight Decay | $1e - 4$ |
| Learning Rate Scheduler | ReduceLROnPlateau (patience: 50) |
| Epochs | 3 |

The models are trained for a maximum of 3 epochs using PyTorch Lightning's `Trainer`, with all computations performed on a single GPU device. The data are publicly available and can be requested at `https://openneuro.org/datasets/ds003020/`. All experiments and models were trained on a server equipped with four NVIDIA A100 GPU cards (80GB RAM each connected through NVLINK) and 2 TB of System RAM.

## B  Comparative Analysis of Generative Decoding and Contrastive Learning Approaches

In this section, we detail the connections and differences between the generative decoding approach described in [19] and our proposed contrastive learning decoder. The aforementioned work employs a Bayesian encoding technique, learning subject-specific encoding and noise models from data, and estimates likelihood probabilities using a large language model (LLM) as a generator of candidate sentences. This enables open-vocabulary text generation guided by fMRI data. In contrast, our

work adopts a more modern and flexible approach based on contrastive learning, aiming to learn cross-subject latent representations of both text and fMRI data.

We will delve into the mathematical foundations of both methods and highlight their similarities and differences.

Let $y$ denote the fMRI data with shape $(t, v)$, where $t$ represents time and $v$ represents the number of voxels. Let $x$ be the downsampled text features extracted by a pretrained language model (GPT-1 for Bayesian decoding and LLaMA3-8B for contrastive decoding), with shape $(t, f)$, where $f$ denotes the number of text features.

The encoding model is defined by a set of learned weights $W$ that maps $x$ to $y$, with $W$ having shape $(f, v)$:

$$\hat{y} = xW$$

In the Bayesian approach, $y$ is modeled as a multivariate Gaussian distribution:

$$y \sim \mathcal{N}(xW, \Sigma)$$

where $\Sigma$ is the covariance matrix of shape $(v, v)$.

Here the goal is to model the posterior distribution $p(x \mid y)$. Applying Bayes' theorem, we can write:

$$p(x \mid y) = \frac{p(y \mid x)p(x)}{p(y)}$$

Since $p(y)$ is constant with respect to $x$, we focus on the numerator $p(y \mid x)p(x)$. The problem thus reduces to estimating a good encoding model (i.e., accurate estimates of $p(y \mid x)$) and utilizing a pretrained language model to estimate the prior probabilities $p(x)$.

The likelihood $p(y \mid x)$ can be expressed as:

$$p(y \mid x) \propto \exp\left(-\frac{1}{2}(y - xW)^\top \Sigma^{-1}(y - xW)\right)$$

Taking the negative logarithm yields the loss function:

$$\mathcal{L} = (y - xW)^\top \Sigma^{-1}(y - xW)$$

Here, the quadratic form represents the residuals between the model predictions and the measurements, re-weighted by the inverse covariance matrix.

Expanding the terms, we obtain:

$$\mathcal{L} = y^\top \Sigma^{-1} y - 2xW\Sigma^{-1}y^\top + xW\Sigma^{-1}W^\top x^\top$$

This results in a $(t, t)$ matrix of residuals. By taking the trace, we obtain a scalar loss function that we can minimize to learn the encoding model weights.

The most significant term is the interaction term:

$$xW\Sigma^{-1}y^\top$$

which involves matrices of shapes:

$$(t, f) \times (\mathbf{f}, \mathbf{v}) \times (\mathbf{v}, \mathbf{v}) \times (v, t)$$

The learnable parameters here are the encoding weights $W$ and potentially the inverse covariance matrix $\Sigma^{-1}$.

In our contrastive model, we employ two learned functions, approximated by neural networks, to map $x$ and $y$ into a shared latent space $z$ of dimensionality $d$:

$$z_x = f(x), \quad z_y = g(y)$$

where $f : \mathbb{R}^f \to \mathbb{R}^d$ and $g : \mathbb{R}^v \to \mathbb{R}^d$.

To simplify the mathematical analysis and highlight the differences and similarities between the two methods, let's assume that both $f$ and $g$ are linear functions:

$$z_x = xA, \quad z_y = yB$$

with $A$ being a matrix of shape $(f, d)$ and $B$ a matrix of shape $(v, d)$, so that $z_x$ and $z_y$ both have shape $(t, d)$.

The objective of the contrastive loss is to make the cosine similarity matrix between $z_x$ and $z_y$ as close as possible to the identity matrix $I$. This can be achieved by computing:

$$S = \frac{z_x}{\|z_x\|} \left( \frac{z_y}{\|z_y\|} \right)^{\top}$$

where $S$ has shape $(t, t)$. We can then use a cross-entropy loss for each element along the diagonal or compute the mean squared error between $S$ and $I$.

By expanding the calculation and ignoring the normalization terms for simplicity, we obtain:

$$S = z_x z_y^{\top} = xAB^{\top}y^{\top}$$

where the matrix multiplications involve shapes:

$$(t, f) \times (\mathbf{f}, \mathbf{d}) \times (\mathbf{d}, \mathbf{v}) \times (v, t)$$

We observe a key connection between the two models: in both cases, we have a similarity (or dissimilarity) matrix of shape $(t, t)$ where the interaction between $x$ and $y$ plays a crucial role.

In the Bayesian model, the encoding projects text features into the brain space and re-weights them based on the noise model, whereas in the contrastive model, this process occurs implicitly through the interaction of the functions $f$ and $g$. In the linear case, the product $AB^{\top}$ takes on a role analogous to $W\Sigma^{-1}$.

However, in the contrastive approach, text features are projected into a latent space (typically with $d \ll v$), resulting in a less descriptive model. This acts as an implicit regularization, but if $d$ is less than the rank of $\Sigma^{-1}$, some information might be lost. This is a suggestion that higher dimensionality plays an important role in brain decoding of language, guiding us in our hyperparameter search.

Thus, while the contrastive model offers greater flexibility—such as training cross-subject models, incorporating nonlinearities, and utilizing compressed latent spaces—it may sacrifice some information about the relationship between $x$ and $y$. A potential future direction could involve modeling $z_y$ directly as a multivariate Gaussian:

$$z_y \sim \mathcal{N}(f(x), Z_{\sigma})$$

where $Z_{\sigma}$ could capture the noise properties of the latent space, possibly modeled by another neural network.

## B.1 Hyperparameter Search

To optimize the performance of our contrastive learning-based model for decoding brain activity, we conducted a hyperparameter search using a random sampling methodology. Our search focused on minimizing the validation loss across a set of 100 randomly sampled configurations from a predefined search space. The hyperparameter sweep was configured as follows:

- **Batch Size (BS)**: We experimented with batch sizes of {512, 1024, 2048}.
- **Learning Rate (lr)**: We explored two learning rates: {1e-4, 1e-5}.
- **Alpha ($\alpha$)**: The weight parameter for the contrastive loss was sampled from {0.5, 0.8}.
- **Temperature ($\tau$)**: We used a fixed value of 0.1 to control the sharpness of the similarity distribution.
- **Loss Function**: Three loss types were tested: {contrastive, mean squared error (MSE), mean contrastive}.
- **Weight Decay (wd)**: Regularization was applied with values {1e-4, 1e-5, 1e-2, 0}.
- **Latent Dimension**: The dimensionality of the shared latent space was varied across {512, 1024, 2048, 4096, 8192, 10000, 16384}.
- **Activation Function**: We tested both {ReLU, Identity} activation functions in the hidden layers.
- **Base Channel Size**: We considered different channel size configurations across layers: {[4096, 2048, 1024], [2048, 1024], [2048]}.
- **Hidden Dimensions**: The hidden layers were configured with varying sizes, including {[2048, 1024, 512], [1024, 512], [1024]}.

The metric used to evaluate the model's performance was the validation loss, which we aimed to minimize. The random sampling method allowed us to explore a diverse set of hyperparameter combinations without performing an exhaustive grid search, which would be computationally expensive.

By systematically varying these key hyperparameters and evaluating each sampled configuration, we were able to identify an optimal combination that balanced both accuracy and generalization across subjects. This hyperparameter search played a crucial role in achieving the strong performance metrics reported in our results.

Accordingly to insights given from comparative analysis we found that linear models with high common and latent dim (equal to the input dimensionality) performed better.

## B.2 Bias, Privacy, and Ethical Considerations

The development of models that decode brain activity into language raises important ethical concerns, particularly regarding bias, privacy, and the responsible use of such technology. One significant concern is the potential for bias in both the models and the data. fMRI datasets, as well as the pre-trained language models used in this study, can inherit biases from the populations they are trained on, which may lead to biased or inaccurate decoding, especially across diverse groups of individuals. This could have serious implications when applying these models in clinical or social contexts.

Another critical issue is privacy. Brain-to-text decoding systems pose a unique risk to neural privacy, as they may allow for the reconstruction of internal thoughts and mental states. This raises questions about consent, data security, and the misuse of brain data in contexts where individuals may not have full control over how their neural activity is used or interpreted. It is essential to develop safeguards to ensure that brain data cannot be used to decode private thoughts without explicit consent, and to explore ways to mitigate any unintended consequences of decoding technologies, such as the risk of surveillance or the exploitation of individuals' cognitive data.

As the field progresses, it will be crucial to establish ethical guidelines that prioritize transparency, fairness, and respect for individual autonomy. Future research should also focus on developing methods to disentangle model biases from genuine cognitive processes and explore the concept of "neural privacy" as a framework for protecting individuals in this emerging area of brain-computer interface technology.

