# OpenReview forum: "Language decoding from human brain activity via contrastive learning"
_NeurIPS.cc/2024/Workshop/UniReps — UniReps_

### Official Review · Reviewer_q8Nb · 2024-10-03
**Novel application; however, the method is not novel and without proper citation; limited data and scope.**

**Rating:** 4
**Confidence:** 4

**Review:**

This paper presents a contrastive learning approach to decode sentences from fMRI brain activity by mapping neural recordings and text embeddings into a shared representational space. While the results appear promising, with reported top-1 accuracy of up to 49.2% and top-10 accuracy of up to 84%, several critical weaknesses must be discussed and addressed.

The primary weakness is that the method itself is not particularly novel. While the authors state that "the core contribution of this work lies in the proposed representation learning pipeline," the approach appears to be a modified version of the contrastive learning framework CLIP introduced by Radford et al. (2021) in their paper "Learning Transferable Visual Models From Natural Language Supervision." The framework presented in Figure 1 reveals that the approach is a relatively straightforward adaptation of existing contrastive learning principles.

Moreover, despite the high similarity between this work and Radford et al.'s framework, the authors neither cite the original work nor adequately address the differences or provide appropriate credit. The authors essentially apply this established technique to brain decoding without significant methodological innovations and without acknowledging the foundational contributions of the earlier work.

Another concern is that the experimental evaluation is notably limited. The results are presented only for three subjects (Table 1), without any comparison to previous work or baseline methods. This lack of comparative analysis makes it impossible to assess the true advancement of their approach over existing techniques in the field. The absence of baselines, ablation studies, or comparative experiments significantly weakens the paper's empirical contribution.

The study relies on data from only three subjects. This limited sample size raises questions about the generalizability of the results and the robustness of the model when applied to a broader population. The authors do not adequately address this limitation or discuss how their findings might scale to larger, more diverse datasets.

In addition, as the authors have pointed out, a key limitation of the proposed approach is that their "decoding is performed through a retrieval module (i.e., sentence identification). This requires access to candidate sentences beforehand, limiting the model’s ability to generalize to brain activity related to sentences that differ significantly from those in the training dataset."

In conclusion, while the paper presents an interesting application of contrastive learning to brain decoding, it represents an incremental advance rather than a significant breakthrough in the field. Future work would benefit from larger sample sizes and more innovative methodological contributions.

Reference:
Alec Radford, Jong Wook Kim, Chris Hallacy, Aditya Ramesh, Gabriel Goh, Sandhini Agarwal, Girish Sastry, Amanda Askell, Pamela Mishkin, Jack Clark, Gretchen Krueger, Ilya Sutskever. Learning transferable visual models from natural language supervision. In International Conference on Machine Learning (ICML 2021).

---

### Official Review · Reviewer_VSLB · 2024-10-05
**The authors propose a method to align fMRI samples with corresponding sentences by training them on the CLIP loss. However, at this stage, the manuscript lacks significant novelty and insight, so I recommend rejection.**

**Rating:** 3
**Confidence:** 4

**Review:**

The authors propose a method to align fMRI samples with corresponding sentences by training them with a contrastive loss. However, at this stage, the manuscript lacks significant novelty and insight, so I recommend rejection.

Strengths
- The paper is easy to understand, and the results indicate that some useful learning does take place.
- Pearson correlation maps show a consistently higher correlation among the same areas, which indicates language-related cortical areas.

Weaknesses                 :
- The novelty of the paper is very limited: it is just one ingredient from the paper BrainCLIP: Bridging Brain and Visual-Linguistic Representation Via CLIP for Generic Natural Visual Stimulus Decoding (https://arxiv.org/abs/2302.12971), where they align text and image samples with fMRI recordings.

---

### Official Review · Reviewer_gZQN · 2024-10-06
**This paper uses contrastive learning to present a cross-subject model for decoding language from fMRI brain activity. While the approach shows some promise in sentence identification, it primarily builds on existing methods without significant novel contributions.**

**Rating:** 4
**Confidence:** 3

**Review:**

Quality:
- The paper provides a detailed description of the model architecture and training process.
- The evaluation metrics (Top-1, Top-10, and Identification Accuracy) are appropriate for the task.
- However, the paper needs a more rigorous comparison with state-of-the-art methods.
- The small sample size (3 subjects) limits the generalizability of the results.
- The paper needs to address potential overfitting concerns adequately.

Clarity:
- The overall structure of the paper is logical and easy to follow.
- The methodology section is detailed, with helpful diagrams and tables.
- The appendix provides comprehensive information on neural network architectures and training setup.

Originality:
- The core idea of using contrastive learning for aligning fMRI and language data is not novel, as similar approaches have been explored in previous works such as BrainCLIP and MindEye.
- The cross-subject aspect of the model is interesting but needs to be sufficiently developed and validated.

Significance:
- The reported performance metrics show improvement over chance levels, but the practical significance of these improvements needs to be well-established.
- The lack of comparison with a broader range of state-of-the-art methods limits assessing the model's true impact.

Pros:
1. Interesting and relevant problem statement.
2. Detailed methodology and appendix enhancing reproducibility.
3. Consideration of ethical implications and privacy concerns.
4. The model does show performance above chance levels.

Cons:
1. The claim of novelty in using contrastive learning for this task is overstated, as similar approaches of aligning into a common shared space via contrastive learning have been used before in the field.
2. The paper needs to sufficiently explore different language models or fMRI preprocessing techniques, limiting the scope of the work.
3. The qualitative comparative analysis with the Bayesian approach is exciting but must demonstrate the proposed method's superiority conclusively through quantification.

---

### Official Review · Reviewer_UyWd · 2024-10-07
**Promising decoding results; can we interpret the representations?**

**Rating:** 6
**Confidence:** 3

**Review:**

The authors train a cross-subject fMRI encoder and show successful sentence identification using a retrieval module. The model significantly outperforms chance levels, suggesting that the learned representations may capture some useful information about how semantic representations are organized in the brain, and how these representations may differ between subjects.

Despite the promising decoding results, there are some shortcomings of the paper. In general, for a 4 page format, there is too much space allocated to Methods and not enough space for the Results. For example, I feel that Figure 2 does not add much to the paper, and that space could have been used for more interesting analyses that actually seek to interpret the learned representation spaces to understand the features that are supporting good retrieval. There are many analyses (dim reduction, clustering of the cosine similarity matrices, etc) that could shed light on the underlying representational dimensions. Punting more methods detail to the Supplement could have created room for these sorts of analyses. As a field, we must not be satisfied with mere prediction scores - the authors' modeling paradigm seems ideally suited to translate good decoding into some new ideas about how the information is actually organized in the brain.

That said, because the methodology appears sound overall, and the paper is well motivated, I will recommend acceptance.  UniReps seems like a good venue for the authors to receive feedback on this work.

---

### Decision · Program_Chairs · 2024-10-10

**Decision:**

Accept

**Comment:**

We would like to begin by thanking the authors and reviewers for their valuable contributions and efforts. In the following, we will provide the rationale behind the decision.

The paper presents a contrastive learning approach for training a cross-subject fMRI encoder to decode sentences, with results indicating performance above chance levels and some potential for understanding brain-based semantic representations. However, reviewers identified several significant weaknesses: the method shows limited novelty, closely following approaches like CLIP and BrainCLIP, and lacks rigorous comparisons to state-of-the-art baselines. The study's small dataset (three subjects) raises concerns about generalizability, and there is limited analysis of the learned representations, reducing the empirical contributions. Additionally, reviewers noted that the claims about the paper's contributions were overstated, and recognition of prior work was insufficient.

Despite these issues, acceptance is suggested because the extended abstract track aims to support early-stage research which are fit in the workshop topics, and presenting at the poster session provides an excellent opportunity for the authors to engage in constructive discussions and refine their work. However, it is strongly recommended that the authors address the reviewers' concern in the camera ready version of the paper,, particularly by recognizing prior literature more clearly and incorporating the suggested feedback, to strengthen the paper’s impact and future development.